# Directly Measuring the Power-law Exponent and Kinetic Energy of Atmospheric Turbulence Using Coherent Doppler Wind Lidar

Jinhong Xian[1,2], Chao Lu[2], Xiaoling Lin[2], Honglong Yang[2], Ning Zhang[1,3], Li Zhang[2]

[1] School of Atmospheric Sciences, Nanjing University, Nanjing 210023, China
[2] Shenzhen National Climate Observatory, Meteorological Bureau of Shenzhen Municipality, Shenzhen 518040, China
[3] Key Laboratory of Urban Meteorology, China Meteorological Administration, Beijing, 100089, China

*Correspondence to*: Honglong Yang (yanghonglong@weather.sz.gov.cn); Ning Zhang
(ningzhang@nju.edu.cn)

**Abstract.** Atmospheric turbulence parameters, such as turbulent kinetic energy and dissipation rate, are of great significance in weather prediction, meteorological disasters, and forecasting. Due to the lack of ideal direct detection methods, traditional structure function methods are mainly based on Kolmogorov's assumption of local isotropic turbulence and the well-known −5/3 power law within the inertial subrange,

which limits their application. Here, we propose a method for directly measuring atmospheric turbulence parameters using coherent Doppler wind lidar, which can directly obtain atmospheric turbulence parameters and vertical structural features, breaking the limitations of traditional methods. The first published spatiotemporal distribution map of the power-law exponent of the inertial subrange is provided in this study, which indicates the heterogeneity of atmospheric turbulence at different altitudes, and also

indicates that the power-law exponent at high altitudes does not fully comply with the −5/3 power law, proving the superiority of our method. We analyze the results under different weather conditions, indicating that the method still holds. The turbulent kinetic energy and power-law index obtained by this method are continuously compared with the results obtained with an ultrasonic anemometer for a month-long period. The results of the two have high consistency and correlation, verifying the accuracy and

applicability of the proposed method. The proposed method has great significance in studying the vertical structural characteristics of atmospheric turbulence.

## 1 Introduction

Turbulence is the main form of motion in the atmospheric boundary layer. It plays a major role in the transportation and exchange of momentum, heat, water vapor, and matter between the surface of the

Earth and the atmosphere, directly affecting human life and production activities and playing a crucial role in atmospheric motion and weather evolution (Byzova et al., 1989; Stull, 1988; Gottschall and Peinke, 2008). At present, there are still many difficulties related to the study of turbulence, and addressing them requires improvements in observational technology. Therefore, the development of atmospheric turbulence detection technology can strengthen our understanding of atmospheric turbulent

motion and accelerate the development of atmospheric turbulence theory. In atmospheric modelling in particular, it is of great significance to obtain the relevant parameters and structural characteristics of atmospheric turbulence (Banakh, 2013).

In the past, there were few detection methods for measuring or inferring low-level turbulence parameters, mainly through the installation of three-dimensional ultrasonic anemometers in meteorological gradient towers (Sathe and Mann, 2013). The detection height, density, and detection ability of the detectors were limited, which hindered the development of boundary layer turbulence theory. Fortunately, with the development of remote sensing technology, as an active remote sensing device with fast response speed and three-dimensional scanning ability, coherent Doppler wind lidar has gradually become the main means of obtaining low-level atmospheric turbulence intensity (Mann et al., 2010; Choukulkar et al., 2017; Bonin et al., 2017; Bonin et al., 2016; Jin et al., 2022; Smalikho et al., 2005; Branlard et al., 2013; Banakh et al., 2021; O'connor et al., 2010).

According to Kolmogorov's theory of local homogeneity and isotropy, the structure function can only be determined by the kinetic energy dissipation rate (Kolmogorov, 1941; Kolmogorov, 1991). Therefore, the structure function can be calculated based on the wind speed fluctuation term, and the kinetic energy dissipation rate can in turn be estimated based on the relationship between the structure function and the kinetic energy dissipation rate given by a statistical turbulence model (Sathe and Mann, 2013). It is called the structure function method, which is an indirect method. Based on this principle, researchers have proposed different acquisition methods. In 2002, Frehlich and Cornman obtained the spatial statistical characteristics of a simulated turbulent velocity field using radial velocity estimation from coherent Doppler lidar data, and subsequently calculated the turbulent energy dissipation rate (Frehlich and Cornman, 2002). In 2005, Smalikho et al. used coherent wind lidar to invert atmospheric turbulence parameters with two methods, i.e., the Doppler spectral width and height structure function, using the range height indicator (RHI) scanning mode. The experimental results were numerically simulated, which confirmed the reliability of these two methods (Smalikho et al., 2005). In 2008, Frehlich and Kelley obtained turbulence parameters in the boundary layer using longitudinal and transverse structure function methods in the plane position indicator (PPI) scanning mode with fixed pitch angle and changing azimuth angle (Frehlich and Kelley, 2008). The inversion results were compared with the detection results of ultrasonic anemometers, and the data consistency was good. In 2012, Chan and Lee divided radial wind field data into subsectors under the PPI scanning mode and calculated the turbulence dissipation rate within each subsector using velocity structure functions, thus obtaining the spatial distribution of the turbulence dissipation rate (Chan and Lee, 2012). In 2017, Smalikho and Banakh used the azimuth structure function in the velocity azimuth display (VAD) scanning mode to invert atmospheric turbulence parameters, and extended the applicability of this method from the convective boundary layer to the stable boundary layer (Smalikho and Banakh, 2017). From 2017 to 2019, Zhai et al. studied the vertical structure characteristics of turbulence in the atmospheric boundary layer using various lidar observation models such as the VAD, PPI, and RHI (Zhai et al., 2017). They analyzed interaction characteristics between a wind turbine wake and atmospheric turbulence under the influence of underlying surfaces with different roughness, and explored the influence of atmospheric turbulence on the evolution process of aircraft wake vortices. In 2023, Wang et al. used shipborne coherent Doppler lidar to measure the energy dissipation rate and wind shear intensity of turbulent flows at sea based on the structure function method, achieving the classification of turbulent mixing sources (Wang et al., 2023). These previous studies are based on the indirect acquisition of atmospheric turbulence parameters

using a structure function, which in turn relies on the assumptions of isotropy and a power-law exponent of –5/3.

Due to the extremely complex turbulence in the atmospheric boundary layer, the assumptions of isotropy and the −5/3 power law are more realistic in the near surface layer of the atmosphere, but may not fully hold true in some complex terrains or high altitudes. Panofsky et al. specifically observed various turbulence structures under complex terrain conditions and found that different terrain conditions have different turbulence energy spectral characteristics (Panofsky et al., 1982). The study by Chellale

et al. also indicates that the characteristics of turbulent energy spectra are influenced by factors such as terrain and spatial location (Chellali et al., 2010). The current methods of measuring atmospheric turbulence parameters using wind lidar data based on Kolmogorov's assumption of local isotropic turbulence and the well-known –5/3 power law are indirect, which have limited applicability and cannot produce accurate turbulence processes. Therefore, the aim of this study is to find a more direct method

for obtaining atmospheric turbulence parameters from the perspective of spectral analysis, without the assumptions of local isotropic turbulence and the −5/3 power law. In Section 2, we introduce the instruments and data quality control methods. In Section 3, we presented the obtained turbulence spectra and propose a method for directly obtaining atmospheric turbulence parameters using coherent Doppler wind lidar data. On this basis, we study the vertical characteristics of atmospheric turbulence obtained

from wind lidar data and compare them with the results from three-dimensional ultrasonic anemometers to verify their accuracy, as shown in Section 4. The main conclusions of this study are presented in Section 5.

**2 Instruments and Data Quality Control**

The Shenzhen Shiyan Observation Base (113.90586 °E, 22.65562 °N) has a 356 m high meteorological

gradient observation tower, which is the tallest in Asia (Zhou et al., 2023). The surrounding terrain and landforms of the observation tower are shown in Figure 1(a), and Figure 1(b) shows the wind lidar instrument beneath the gradient observation tower. Four three-dimensional ultrasonic anemometers (CSAT3, Campbell Scientific, Utah, USA) are installed at heights of 10 m, 40 m, 160 m, and 320 m on the gradient observation tower. From Figure 1, it can be seen that there are no large obstacles (such as

tall buildings) around the tower. Located 1–2 km northeast of the tower there is farmland, while tall buildings are located in distant suburbs. The terrain to the south and northwest of the gradient tower is generally flat, almost completely covered by forests and lakes. Due to its excellent geographical location with no obstructions around it, the monitored data are highly representative. We used wind lidar data and data obtained with the three-dimensional ultrasonic anemometer in this study. The observational

frequency of the ultrasonic anemometer is 10 Hz, and the wind speed accuracy can reach 0.1 m/s. The wind lidar (DSL-W, Darsunlaser Technology Co., Ltd., Shenzhen, China) detects a blind spot of 30 m, with a maximum detection height of 3 km and a vertical resolution of 30 m. Its time resolution is 5 s, which means its observational frequency is 0.2 Hz. Specific performance indicators of the wind lidar instrument and three-dimensional ultrasonic anemometer are shown in Table 1.

The values of turbulence parameters are extremely dependent on the precision of wind speed measurements, using obtained as a time series, and the presence of too many abnormal signals can lead

to the inference of abnormal turbulence parameters. Therefore, it is necessary to conduct data quality control to ensure the reliability of the observed data. For the wind lidar data, an overall inspection was conducted on the wind speed measurements every 30 min every day, eliminating data with wind speed measurements deviating from the average by more than three standard deviations (Qiu et al., 2023). For a single wind speed profile data set, if more than 20% of the data points were lost below 500 m, the entire profile was discarded. For the three-dimensional ultrasonic wind speed data, we calculated the average and standard deviation of the observed values within 30 min, and any observed values deviating from the average by more than three times the standard deviation were marked as abnormal data and assigned as missing values. We repeated the above vetting process three times. During processing, data interpolation was not performed. If the number of missing measurements exceeded 20% within 30 min, the data were discarded. In order to eliminate wind speed errors caused by an installation tilt error of the ultrasonic anemometer, it was necessary to rotate the coordinates of the wind speed; here, we corrected the coordinate axis using the double rotation method (Zhou et al., 2023).

The analysis of atmospheric boundary layer turbulence fluctuations, such as correlation analysis and spectral analysis, is based on the assumption that atmospheric turbulence fluctuations are stationary. The actual atmospheric turbulence field is influenced by various factors and does not have stationarity characteristics (Massman, 2006). However, if a shorter observation time is used, under relatively stable weather conditions and flat underlying surface conditions, atmospheric turbulence can be approximated as static. Turbulence stationarity requires that the main statistical variables of turbulence remain stable within the observation time, that is, the mean of the variance of the entire time period within an observation time period is roughly equal to the mean of the sum of variances of each period (Massman, 2006). In this study, data screening is conducted by determining whether the deviation between the mean variance within a 30 min observation time and the mean variance of six 5 min covariance samples within the same period is less than 0.3. Turbulence stationarity can be achieved through the stationarity coefficient, $\Delta st$, as shown in Equation (1) (Massman, 2006)

$$\Delta st = \left|(\sigma^5 - \sigma^{30})\right|/\sigma^{30}, \tag{1}$$

where $\sigma^{30}$ represents the variance of wind speed within 30 min and $\sigma^5$ represents the average variance of six 5 min wind speed measurements made within 30 min. If $\Delta st > 0.3$, the data within these 30 minutes are discarded. The ratio of discarded data to all data used in this study is approximately 10%.

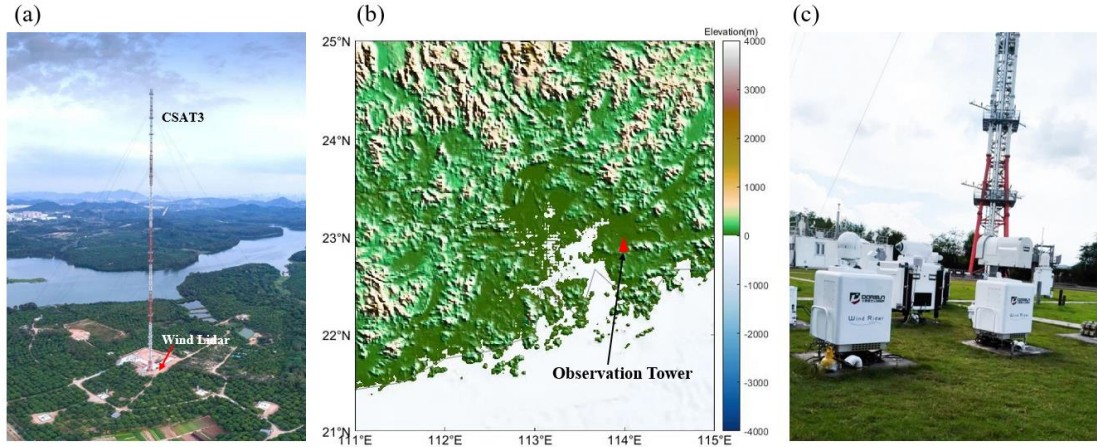

**Figure 1.** Layout diagram (a), topographic map (b) of the surrounding area of the meteorological gradient observation tower, and installation diagram of the wind lidar below the tower (c).

  **Table 1.** Performance parameters of the ultrasonic anemometer and wind lidar instrument

| | Metrics | Technical Performance Requirements |
|---|---|---|
| Ultrasonic anemometer | Observational frequency | 10 Hz |
| | Resolution of the wind speed | ≤0.1 m/s |
| | Resolution of the wind direction | ≤1° |
| | Range of wind speed measurements | 0–40 m/s |
| Wind Lidar | Minimum detection altitude | ≤30 m |
| | Maximum detection altitude | 3 km |
| | Distance resolution | 30 m |
| | Observational frequency of the wind profile | 0.2 Hz |
| | Resolution of the wind speed | ≤0.1 m/s |
| | Resolution of the wind direction | ≤1° |
| | Range of wind speed measurements | 0–60 m/s |
| | Range of wind direction measurements | 0°– 360° |

## 3 Methodology

### 3.1 Theory

The three-dimensional wind speed measured by the wind lidar and ultrasonic anemometers can be represented as $U(z, t)$, $V(z, t)$, and $W(z, t)$; where $U$ is the east–west direction, $V$ is the north–south
155  direction, $W$ is the vertical direction, $z$ is the height, and $t$ is the time. Fast Fourier transform (FFT) calculations are performed on the wind speed within a certain time range at a certain height, $z$, to obtain the turbulence power spectrum, $S$, as

$$S_z(f) = \alpha \varepsilon^{-\frac{2}{3}} f^n ,\tag{2}$$

where $f$ is the frequency, $\alpha$ is the Kolmogorov constant, $\varepsilon$ is the dissipation rate, and $n$ is the power-law
160  exponent. It is generally believed that $n = -5/3$ (Kolmogorov, 1991). In this study, it is considered a variable.

By taking the logarithm of both sides of Equation (2) we get

$$\log(S_z(f)) = \log(\alpha\varepsilon^{-\frac{2}{3}}) + n\log(f). \tag{3}$$

According to Equation (3), we set

$$x = \log(f), \tag{4}$$

$$b = \log(\alpha\varepsilon^{-\frac{2}{3}}), \tag{5}$$

which are substituted into Equation (3) to yield

$$\log(S_z(f)) = b + nx. \tag{6}$$

The slope can be obtained by performing linear fitting on $x$ and $\log(S_z(f))$, which yields the power-law exponent $n$. The turbulent kinetic energy, $\kappa$, in a certain frequency range $[f_0, f_1]$ can be obtained by

$$\kappa = \sum_{f_0}^{f_1} S_z(f). \tag{7}$$

When $\kappa$ and $n$ are known, the dissipation rate can be obtained from Equation (2). Therefore, this paper mainly discusses turbulent kinetic energy, $\kappa$, and the power-law exponent, $n$.

### 3.2 Method

Compared with traditional structure function methods that rely on the assumption of isotropy and the $-5/3$ power law, we propose a method based on spectral analysis to obtain the turbulence parameters directly. From the perspective of spectral analysis, it is possible to have a simpler and clearer understanding of atmospheric motion at different scales and types, with wider applicability and more representative significance. The scale range of vortex motion in the atmosphere is very wide, making it difficult to obtain a full frequency spectral distribution function. Usually, based on the nature of the problem being studied, the required scale range and total sampling time of the corresponding turbulent vortices are determined. For example, when studying the contribution of thermal convection or gravitational internal waves to energy in the boundary layer atmosphere, the total sampling time cannot be less than 1 h, as detection data are required to reflect vortex motion on a timescale of 10 min (Zeng et al., 2010). When studying the contribution of turbulence to energy in the inertial subrange, the total sampling time usually takes 20 min, and the corresponding vortex scale ranges from a few seconds to several tens of seconds. In discrete FFT, $2^N$ data points are required, where $N$ is an integer. Therefore, in this paper, we conducted FFT calculations for $2^8 \times 5$ s points (i.e., approximately 20 min) using the data obtained from the wind lidar and $2^{14} \times 0.1$ s points (i.e., approximately 27 min) for the data obtained with the ultrasonic anemometer. According to the Nyquist sampling law, the highest frequencies of turbulence spectra that can be monitored by ultrasonic anemometers and wind lidar are 5 Hz and 0.1 Hz, respectively.

We compared the wind speed data of the wind lidar at a height of 330 m with the ultrasonic anemometer data at a height of 320 m, which are within the 30 m resolution of the wind lidar data. Figures 2(a)–(c) show a comparison of turbulence spectra in the three directions ($U$, $V$, $W$) obtained with the wind lidar and ultrasonic anemometer. From the graph we can see that in all three directions the spectra have high consistency within the overlapping frequency range of $10^{-2.5}$ to $10^{-1}$ Hz and are in good compliance with the $-5/3$ power-law index. Figures 2(d)–(f) provide the correlation coefficients (R)

corresponding to Figures 2(a)–(c), respectively. It can be seen that the correlation coefficients are greater than 0.9 in all three directions. This proves that wind lidar can effectively monitor the turbulence spectrum of wind within the frequency range of $10^{-2.5}$ to $10^{-1}$ Hz. In this study, in order to avoid differences caused by the size of the frequency domain, the frequency range was selected as the overlapping area of the two during the comparison process. Based on the method proposed in Section 3.1, the turbulent kinetic energy and power-law exponent within this frequency range were obtained. The structure function method assumes isotropy in atmospheric turbulence and cannot obtain power-law exponents. By using wind lidar to obtain atmospheric turbulence spectra, not only can the turbulent kinetic energy be directly obtained, but also the power-law exponents, thus making the spectra applicable to different atmospheric conditions. We verify the applicability and accuracy of this method in the next section.

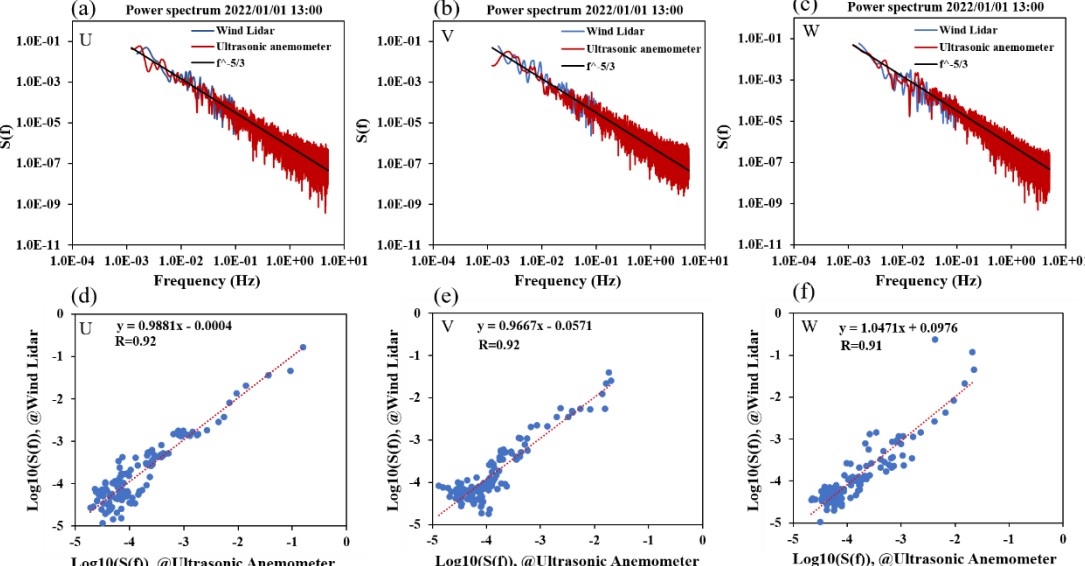

**Figure 2.** Comparison of the turbulence spectra obtained with the wind lidar and the ultrasonic anemometer in three directions: (a) *U*, (b) *V*, and (c) *W*, and the corresponding correlations (d), (e), and (f).

## 4 Results and Discussion

### 4.1 Detection Performance under Weak Convection

Using the method proposed in Section 3.2, based on the high-resolution wind lidar data, we obtained the kinetic energy and power-law exponent at different heights; here a height of 330 m was selected as an example and the results are compared with those obtained with the three-dimensional ultrasonic anemometer. As mentioned in Section 3.2, in all subsequent comparisons, the time resolutions of the wind lidar and ultrasonic anemometer data were maintained at 20 min and 27 min, respectively. Figure 3 shows a comparison of the turbulent kinetic energy obtained by wind lidar and the three-dimensional ultrasonic anemometer on January 1, 2022. The weather on that date was clear and cloudless, with an average temperature of 14 ℃. The main wind direction was easterly, with an average horizontal wind speed of ~3 m/s. From the graph, it can be seen that the results of both in all three directions are relatively consistent, verifying the accuracy of wind lidar in monitoring the vertical characteristics of atmospheric turbulence.

On this basis, we produced spatiotemporal distribution maps of the turbulent kinetic energy and power-law exponent, as shown in Figure 4, where panels (a)–(c) correspond to wind speed components *U*, *V*, and *W*, respectively. In our model we do not assume the −5/3 power law, but instead allow it to be a free parameter. Thus, we also present a spatiotemporal distribution map of the power-law exponent of the inertial subrange in the three directions in Figures 4(d)–(f). We can observe the vertical structure and

characteristic of turbulence in Figure 4. From Figures 4(a)–(c), it can be seen that the kinetic energy in the *U* and *V* directions was relatively consistent near the ground surface, but significant differences are seen at high altitudes, such as between 18:00 and 24:00 (local time). The turbulent kinetic energy in the *W* direction was relatively small because compared to the horizontal wind speed, the vertical wind speed was relatively small. From Figures 4(d)–(f) it can be seen that the power-law exponent in the *U* and *V*

directions was relatively consistent most of the time, which is more in line with isotropic theory. However, at high altitudes at night there were significant differences, where the power-law exponent in the vertical direction (*W*) showed a phenomenon of being high at night and low during the day. This is because the vertical wind speed is mainly driven by ground heating radiation during the day. At night, due to the weakening of solar radiation, the kinetic energy in the vertical direction is suppressed and the power-law

exponent increases.

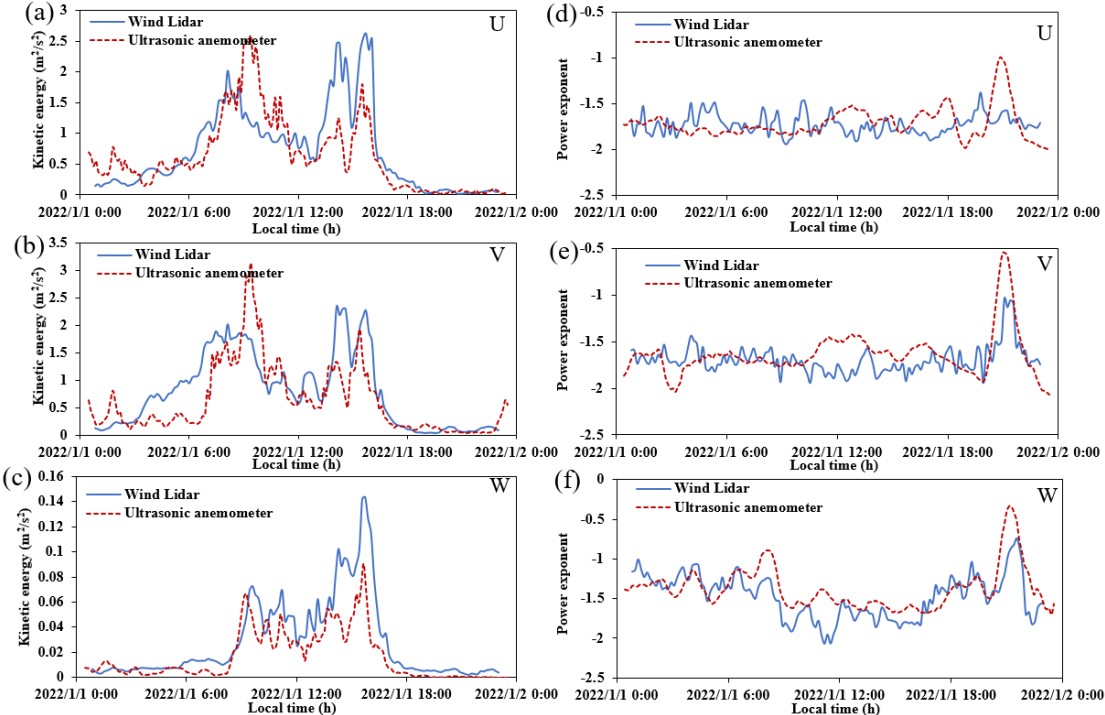

**Figure 3.** Comparison of the turbulent kinetic energy obtained from the wind lidar and three-dimensional ultrasonic anemometer on **January 1, 2022** in the (a) *U*, (b) *V*, and (c) *W* directions and the power-law exponent distribution
in the (d) *U*, (e) *V*, and (f) *W* directions.

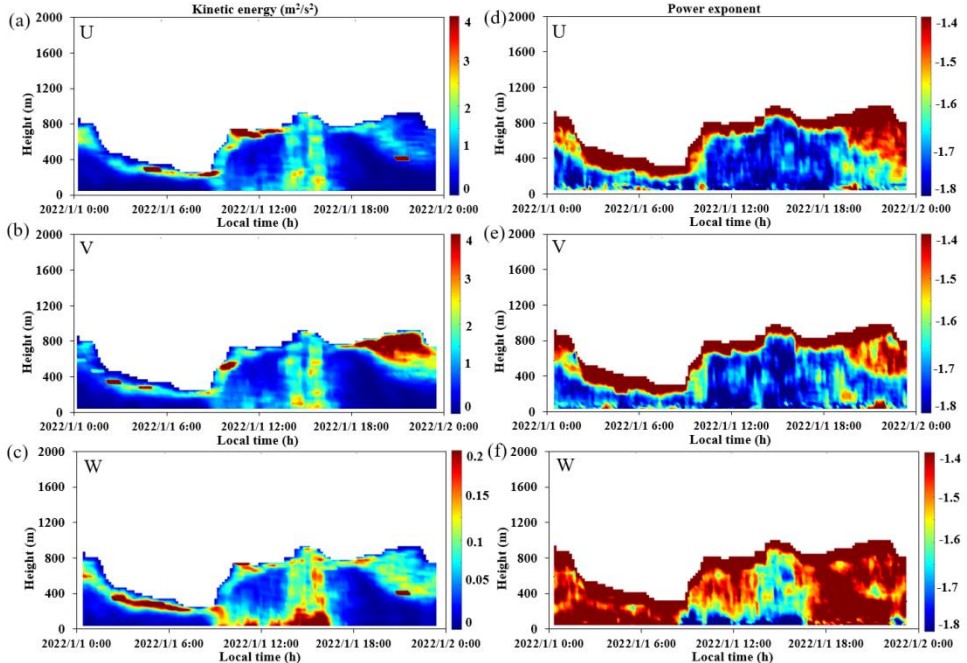

**Figure 4.** Temporal and spatial variations of the turbulent kinetic energy obtained by wind lidar on January 1, 2022 in the (a) *U*, (b) *V*, and (c) *W* directions and the power-law exponent distribution in the (d) *U*, (e) *V*, and (f) *W* directions.

Furthermore, we present the derived power-law exponent profile in the three different directions at different local times on January 1, 2022 in Figure 5. From the graph it can be seen that the power-law exponent changed similarly in the *U* and *V* directions with height. From Figure 5(a) it can be seen that at altitudes ~600–800 m, the turbulence in all three directions conformed to the assumptions of homogeneity and isotropy, while at other altitudes, the turbulence in all three directions was anisotropic.

As the height increased, the power-law exponents in all three directions increased, corresponding to the power-law exponent distributions shown in Figures 4(d)–(f), where there was a layer with a higher power-law exponent, indicating the presence of kinetic energy suppression at the top of the boundary layer. Both Figure 4 and Figure 5 intuitively and clearly reflect that the atmosphere was not isotropic and the power-law exponent was not entirely −5/3, indicating that the atmospheric turbulence parameters

obtained by the spectral analysis method proposed in this study are more in line with the actual atmospheric conditions and therefore have good accuracy.

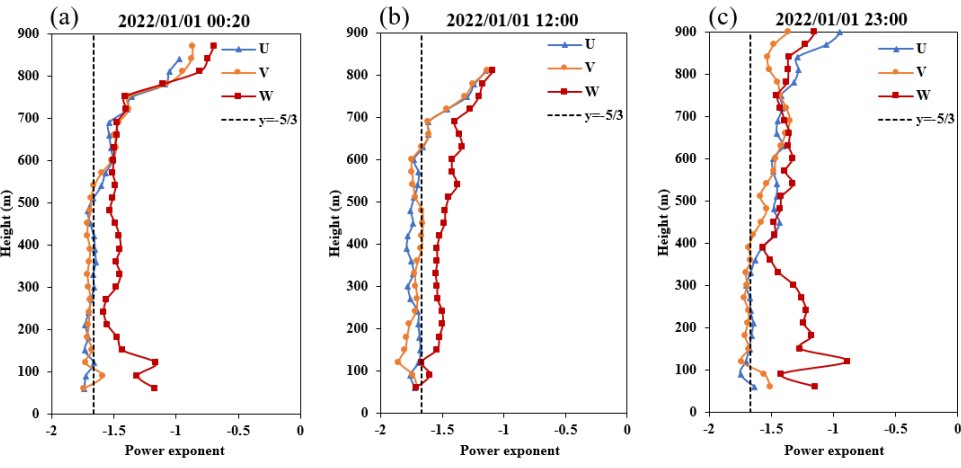

**Figure 5.** Power-law exponent profile of the wind speed in different directions with altitude on January 1, 2022 at (a) 00:20, (b) 12:00, and (c) 23:00 local time.

### 4.2 Detection Performance under Strong Convection

To verify the effectiveness of the proposed method under different weather conditions, Figure 6 shows the spatiotemporal variations of the turbulent kinetic energy and the power-law exponent on September 11, 2022. The weather on this day was clear and cloudless, with an average temperature of 27 ℃. The main wind direction was southwest, with an average wind speed of about ~3 m/s. From the graph it can be seen that the convection was stronger on this day, with a boundary layer height of about 2000 m. The turbulent kinetic energy in the *U* and *V* directions exhibited significant differences at different altitudes. The power-law exponent was relatively consistent most of the time, more in line with the assumption of isotropy, but exhibited significant differences near the ground at night. This may be due to daytime heating radiation resulting in more homogeneous atmospheric mixing, which is more in line with the isotropic hypothesis. The power-law exponent in the vertical direction (*W*) also showed a phenomenon of high at night and low during the day. Similarly, the turbulent kinetic energy and power-law exponent at a height of 330 m were selected and compared with the results from the three-dimensional ultrasonic anemometer, as shown in Figure 7. The installation of the ultrasonic anemometer on the north side of the gradient tower was likely affected by wind obstruction caused by the tower, with the effect being most significant from 0:00 to 6:00. We infer this obstruction effect based on nighttime data collected by the ultrasonic anemometer, such as near midnight, where the measured turbulent kinetic energy was found to be close to its intensity measured during the day. From the graph, it can be seen that the results of both in all three directions are relatively consistent, further verifying the accuracy and applicability of wind lidar in monitoring the vertical characteristics of atmospheric turbulence.

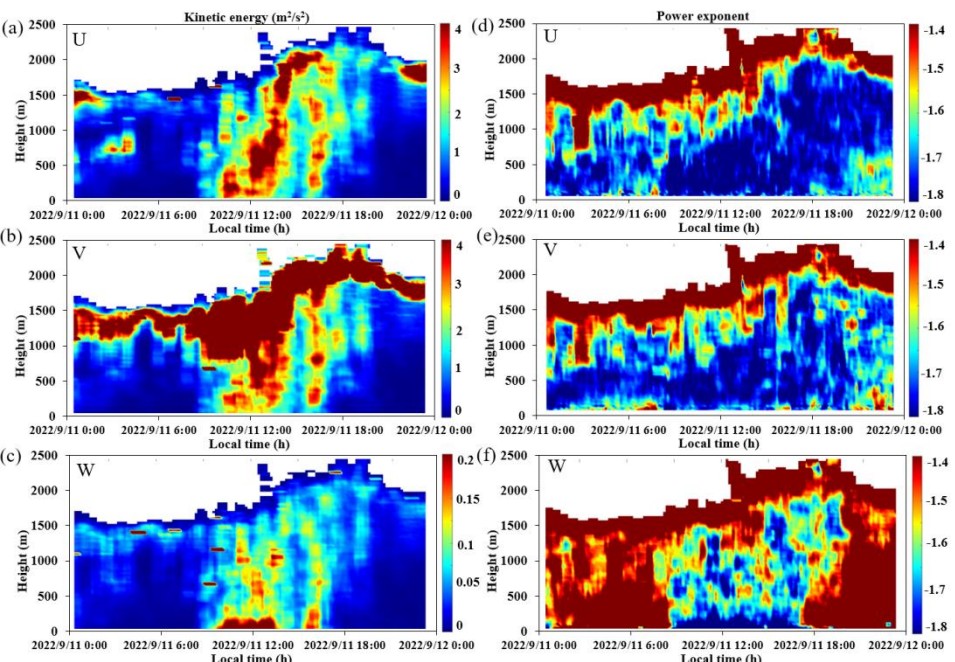

**Figure 6.** Temporal and spatial variations of the turbulent kinetic energy obtained by wind lidar on September 11, 2022 in the (a) *U*, (b) *V*, and (c) *W* directions and the power-law exponent distribution in the (d) *U*, (e) *V*, and (f) *W* directions.

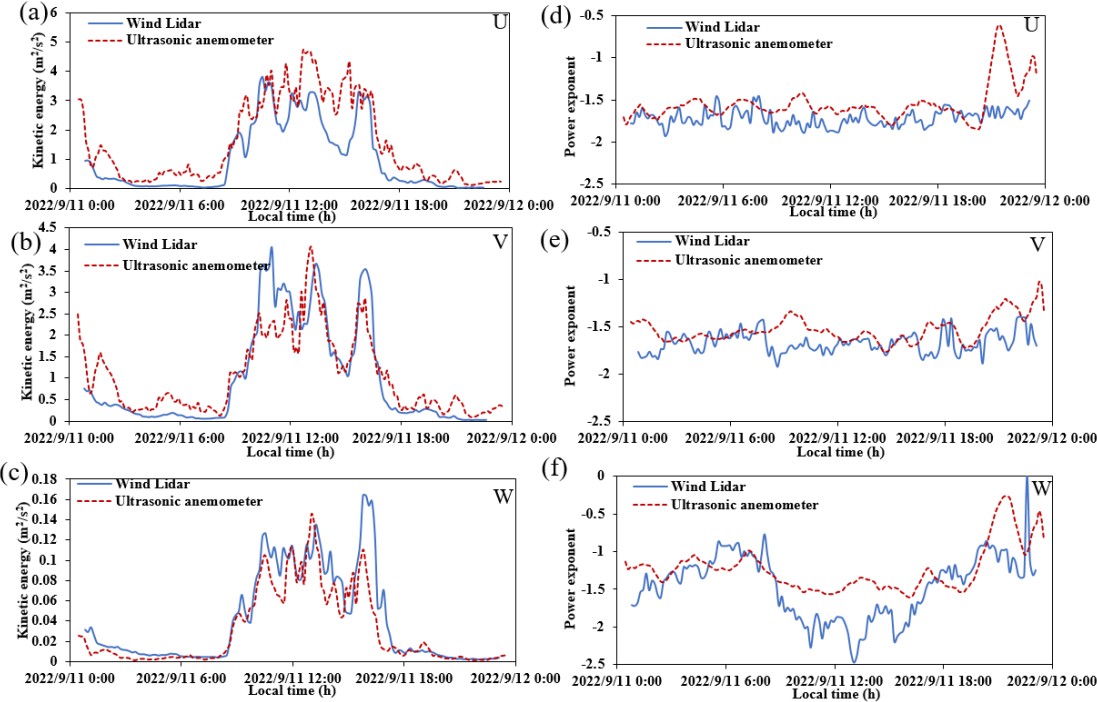


**Figure 7.** Comparison of the turbulent kinetic energy obtained by the wind lidar and three-dimensional ultrasonic anemometer on September 11, 2022 in the (a) *U*, (b) *V*, and (c) *W* directions and the power-law exponent distribution in the (d) *U*, (e) *V*, and (f) *W* directions.

**4.3 Detection Performance in Cloudy Weather**

Figure 8 shows the spatiotemporal variations of the turbulent kinetic energy and power-law exponent on January 14, 2022. On that day, the weather was cloudy with an average temperature of 13 ℃. The main wind direction was easterly, with an average wind speed of 6 m/s. Similar to the previous results, the power-law exponent in the *U* and *V* directions was relatively consistent, in line with the assumption of isotropy. The power-law exponent in the vertical direction (*W*) was mainly affected by

surface temperature radiation. Due to the cover of clouds during the day, the difference in the surface temperature radiation between day and night was not significant, so the phenomenon of a higher power-law exponent at night and a lower exponent during the day was not obvious. Figure 9 shows a comparison of the turbulent kinetic energy obtained by wind lidar and the three-dimensional ultrasonic anemometer on the same date. From Figures 9(a)–(c) it can be seen that the turbulent kinetic energy peaked multiple

times, and the results from the two methods have high consistency under complex weather conditions. In Figures 9(d) and (e), the power-law exponents of the two in the *U* and *V* directions are also relatively consistent. This means that the method proposed has good performance in predicting the turbulent kinetic energy and power-law exponent in these directions in cloudy weather. In Figure 9(f), there is a significant difference in the power-law exponent between the two in the *W* direction. Because it was a cloudy day,

the vertical wind speed on that day was relatively low, about 0.1 m/s. We attribute this difference to the different measurement principles of the different instruments; namely that the ultrasonic anemometer requires a minimum wind speed (~0.01 m/s) to start making measurements, whereas the wind lidar has no such dependency. This indicates that the method proposed in this study can be utilized under different weather conditions and has high applicability.

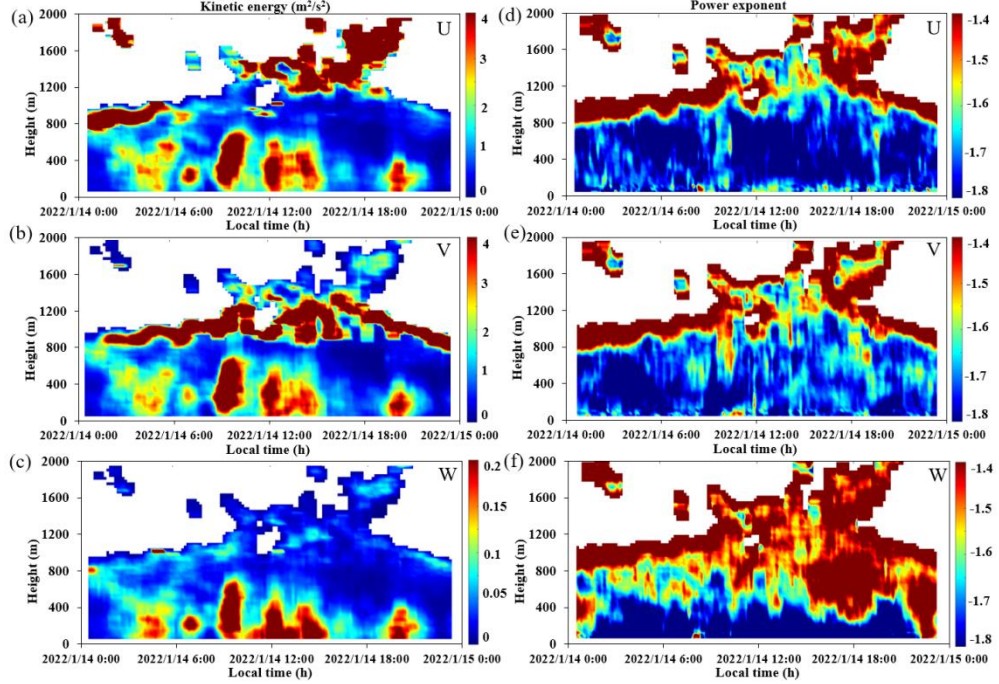

**Figure 8.** Temporal and spatial variations of the turbulent kinetic energy obtained by wind lidar on January 14, 2022 in the (a) *U*, (b) *V*, and (c) *W* directions and the power-law exponent distribution in the (d) *U*, (e) *V*, and (f) *W* directions.

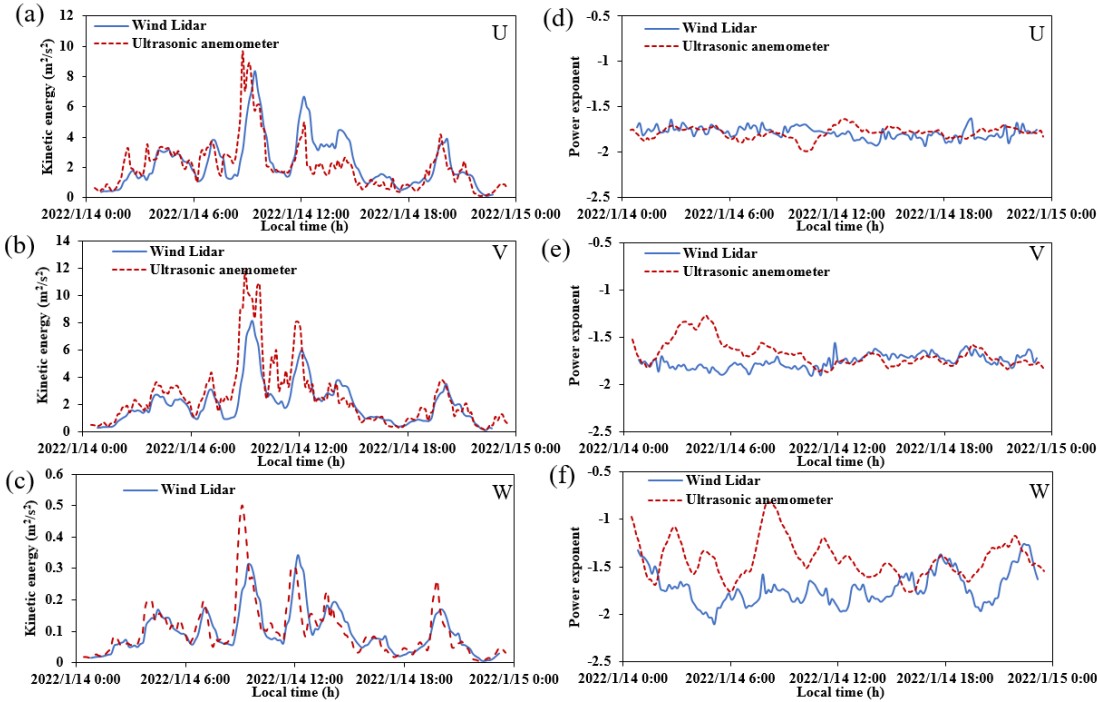

**Figure 9.** Comparison of turbulent kinetic energy obtained by the wind lidar and three-dimensional ultrasonic anemometer on January 14, 2022 in the (a) *U*, (b) *V*, and (c) *W* directions and the power-law exponent distribution in the (d) *U*, (e) *V*, and (f) *W* directions.

### 4.4 Continuous Long-term Verification

Figure 10 shows a comparison of the long-term, continuous turbulent kinetic energy obtained from

the wind lidar and three-dimensional ultrasonic anemometer from January 2nd to 10th, 2022, where panel

(a)– (c) and (d)– (f) correspond to the heights of 160 m and 320 m, respectively. From the graph, we can

see that the results of the wind lidar and ultrasonic anemometer are relatively consistent at the different

heights. Figure 11 shows the correlation of the turbulent kinetic energy measured by the wind lidar and ultrasonic anemometer in all three directions, where panels (a)–(c) and (d)–(f) correspond to the same

heights of 160 m and 320 m, respectively, with sample sizes (N) all greater than 2800. From the figure it can be seen that the results are relatively linear, with correlation coefficients greater than 0.9 and the slope of the fitting line approaches 1 in the *U*, *V*, and *W* directions. Compared to the *U* and *V* components, the consistency of the comparison results in the *W* direction is the highest. This is because under different horizontal wind directions, the results of the *U* and *V* components are more susceptible to interference

from the gradient tower itself, while the vertical wind speed (*W*) is not affected by this, thus yielding better results. This comparison of the continuous results indicates that the proposed method has a high applicability and accuracy.

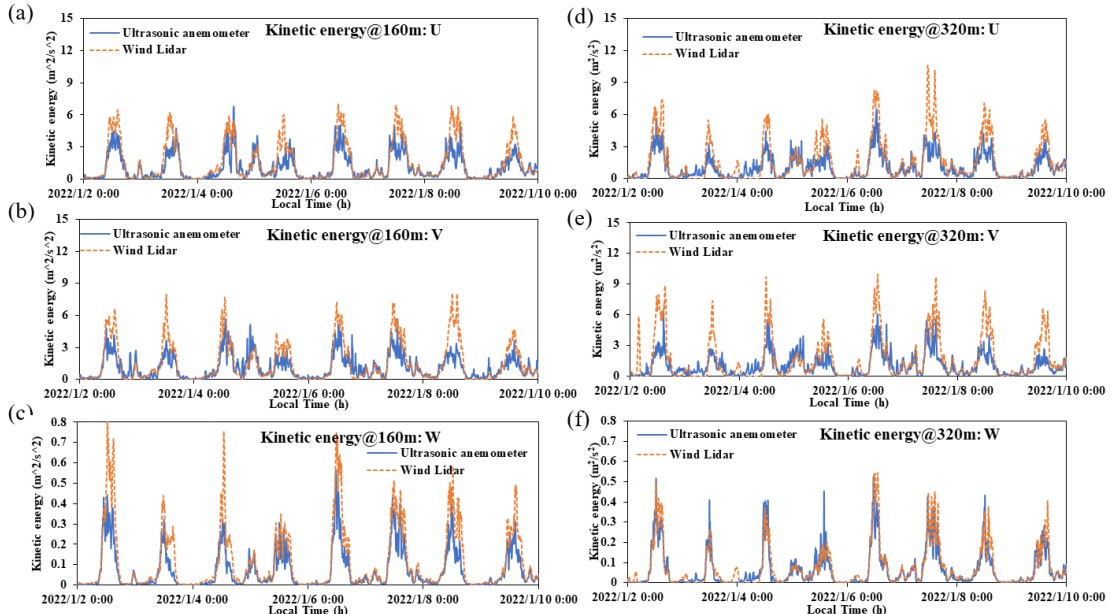

**Figure 10.** Comparison of the turbulent kinetic energy obtained from the wind lidar and three-dimensional ultrasonic
anemometer from January 2nd to 10th, 2022 in the *U*, *V*, and *W* directions at the heights of 160 m (a)–(c) and 320 m (d)–(f).

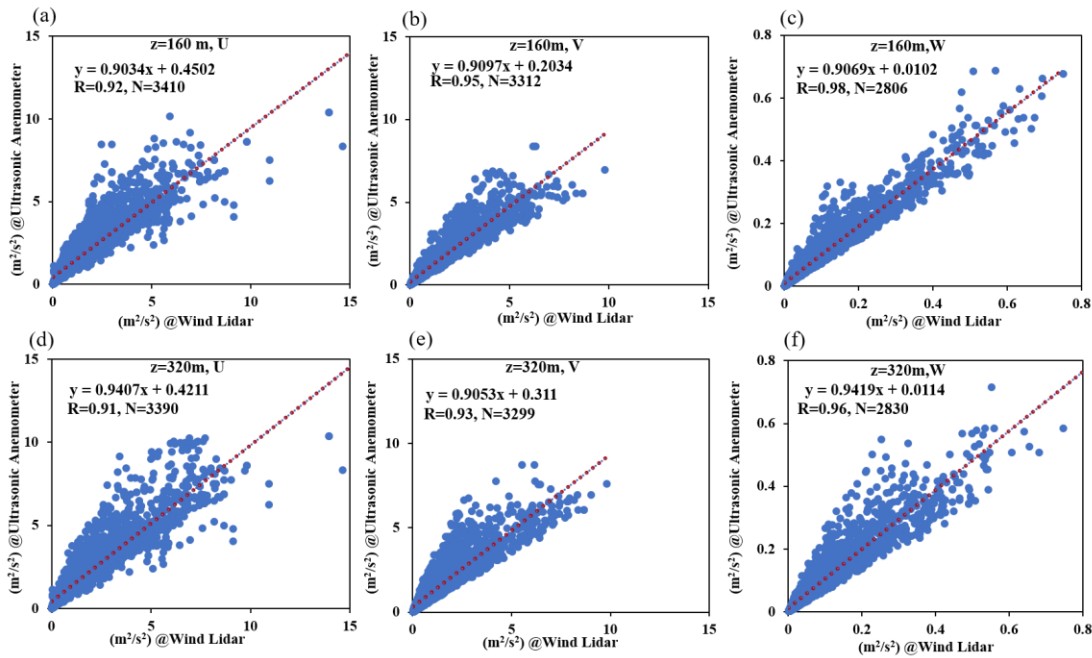

**Figure 11.** Correlation between the turbulent kinetic energy obtained with the wind lidar and three-dimensional ultrasonic anemometer in the *U*, *V*, and *W* directions at the heights of 160 m (a)–(c) and 320 m (d-f).

## 5 Conclusion

We proposed a method for directly measuring atmospheric turbulence parameters using coherent Doppler wind lidar from the perspective of spectral analysis without assuming isotropy and the −5/3 power law. The method can intuitively reflect the vertical characteristics of atmospheric turbulence. In our method, the power-law index is left as a free parameter, and based on our results we have presented, for the first time, a power-law exponent spatiotemporal distribution map for the inertial subrange is provided. The results indicate that the atmosphere does not always conform to the assumption of isotropy and the −5/3 power law, which verifies that the proposed method of directly obtaining atmospheric turbulence parameters from spectral analysis is more in line with the actual atmospheric conditions and has good accuracy. The obtained turbulent kinetic energy and power-law exponent were compared with the results obtained with ultrasonic anemometers for different weather conditions and over a long period (one month). The results of the three wind speed components (*U*, *V*, and *W*) showed good consistency, with correlations reaching 0.91, 0.93, and 0.96, respectively. In complex weather conditions, there was still a high degree of consistency between the two results, indicating that the method proposed in this study has high applicability and accuracy. However, the proposed method has some limitations. First, the method based on a spectral analysis that requires the atmospheric turbulence fluctuations to be stable. Next, wind lidar cannot operate during heavy rainfall or snowy weather conditions, so it cannot be guaranteed to be applicable at all times. In addition, due to the maximum observation frequency of 0.2 Hz for wind lidar, the observed scale range of vortex motion is limited. However, the results of this study also indicate that in the inertial subrange, turbulence spectra outside the frequency of 0.2 Hz can be obtained through fitting and extrapolation. Compared with traditional structure function methods that rely on isotropic assumptions and the −5/3 power law, the proposed method has higher applicability and accuracy with

fewer assumptions. It can obtain the spatiotemporal distributions of atmospheric turbulence parameters, which have important significance in weather prediction, meteorological disasters, and forecasting.

**Data availability**

The data are available from the authors upon request.

**Author contributions**

Conceptualization, J.X.; methodology, J.X.; software, J.X.; validation, C.L.; formal analysis, J.X.; investigation, X.L.; resources, C.L.; data curation, L.Z.; writing—original draft preparation, J.X.; writing—review and editing, H.Y. and N.Z.; visualization, X.L.; supervision, H.Y. and N.Z.; project
administration, H.Y. and N.Z.; funding acquisition, H.Y. and N.Z.

**Competing interests**

The contact author has declared that none of the authors has any competing interests.

**Acknowledgements**

We thank Shenzhen Darsunlaser Technology Co., Ltd.

**Financial support**

This work was supported by  the National Key R&D Program of China (2019YFEQ124800),Special Project for Sustainable Development of Shenzhen (KCXFZ202001221173412035), the National Natural Science Foundation of China (NSFC) (42275065,U2342221), Guangdong Province Science and Technology Department Project (2021B1212050024), Scientific research projects of Guangdong
Provincial Meteorological Bureau (GRMC2020M29), Science and Technology Innovation Team Plan of Guangdong Meteorological Bureau (GRM-CTD202003).

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
