# Peer review of "Directly Measuring the Power-law Exponent and Kinetic Energy of Atmospheric Turbulence Using Coherent Doppler Wind Lidar"

_Atmospheric Measurement Techniques, 2023_

## Referee Comment (RC1)

A new method for directly measuring atmospheric turbulence parameters based on coherent Doppler wind LiDAR is proposed in the preprint manuscript. The spatiotemporal distribution map of the power-law exponent of the inertial subrange is published, which indicates the heterogeneity of atmospheric turbulence at different altitudes, and also indicates that the power-law exponent at high altitudes does not fully comply with the −5/3 power law. the detection results under clear sky turbulence and cloudy weather are compared. The detection results were compared under clear and cloudy weather, respectively. I was very interested in this method and the results of the detection, but when I carefully reviewed the manuscript, I found that some issues needed to be resolved before they could be published in the journal. Therefore, my opinion is that major modifications are needed, and the specific suggestions are as follows:

1. In the introduction section, the development process of turbulence detection was introduced, but the differences between these development processes and this manuscript were not discussed. Please carefully refine them. For example, the methods used by Doppler LiDAR to detect turbulence, as described in references 18-22, and their differences and connections with this manuscript.

2. In section 2 of the manuscript, did you mention that your CSAT data is averaged over 30 minutes? Does this exceed the atmospheric turbulence freezing time?

3. In the abstract section, it was explained that you have for the first published a spatiotemporal distribution map of the power law exponent of the inertial subrange, which shows the non-uniformity of atmospheric turbulence at different altitudes. I am not very clear about what I have read in sections 3.1 and 3.2 of your manuscript, because the new method you proposed is not expressed clearly. I can roughly understand it as using spectral analysis for fitting and formulating a system of equations to solve turbulent kinetic energy $\kappa$ and the power-law exponent n (and n is not fixed -5/3). So it is important for you to carefully explain your new method and its differences from previous research methods.

4. Turbulence has both spatial and temporal scales, and directly measuring the spatiotemporal distribution of turbulence has always been an important issue. And we know that most turbulence spectra are distributed between 0.1Hz and 100Hz. In section 3.2 of the manuscript, the highest frequencies of turbulence spectra that can be monitored by ultrasonic anemometers and wind LiDAR are 5 Hz and 0.1 Hz, respectively. Is this the reason for the significant difference in detection results between the two sensors? What is the contribution to the difference in detection results?

5. From your results, the comparison between LiDAR and CSAT is a single point comparison around 320 meters. How can the comparison of single points be close to prove that the spatial distribution of the results detected by LiDAR is correct? Why didn't you compare a few more points below 320 meters in the vertical direction?

6. In the results and discussion of section 4, we believe that its detection ability will be limited for such a system and sampling frequency. For example, whether the consistency of turbulence is better only in clear skies (Figure 3a-c), and the consistency of results becomes worse in severe convective weather (Figure 7a-c)

7. In Figure 9, we can clearly see that when the turbulent kinetic energy is high, the turbulent kinetic energy and power-law exponent of LiDAR and CSAT3 in the vertical direction are significantly different. Why is this? Does it mean that this method is flawed in cloudy weather conditions?

8. Some normative issues in drawing, such as: the area near the ground pointed by the red arrow in Figure 1a is white, while the area where the LiDAR is placed in the right image is clearly a green lawn. This is very confusing; Can the lines in Figures 3, 5, 7, 9, and 10 be distinguished by dotted lines, dashed lines, etc?

9. At the end of section 3.1, turbulent kinetic energy $\kappa$ was introduced. However, there is no mathematical relationship between it and the aforementioned formula, which is very confusing.

---

## Author Comment (AC1)

AMT-2023-249- Response Letter
response letter
en

**AMT-2023-249- Response Letter**

Dear Editor and reviewers,

We would like to thank the reviewers and editor for their comments that have allowed us to further clarify some aspects of the manuscript in this revised version. Hereafter, we report reviewers' comments and our replies (*in italics*). For yours and reviewers' convenience we have put the corresponding major changes introduced in red color in the revised version of the manuscript.

**Reviewer 1:**

A new method for directly measuring atmospheric turbulence parameters based on coherent Doppler wind LiDAR is proposed in the preprint manuscript. The spatiotemporal distribution map of the power-law exponent of the inertial subrange is published, which indicates the heterogeneity of atmospheric turbulence at different altitudes, and also indicates that the power-law exponent at high altitudes does not fully comply with the −5/3 power law. the detection results under clear sky turbulence and cloudy weather are compared. The detection results were compared under clear and cloudy weather, respectively. I was very interested in this method and the results of the detection, but when I carefully reviewed the manuscript, I found that some issues needed to be resolved before they could be published in the journal. Therefore, my opinion is that major modifications are needed, and the specific suggestions are as follows:

**1. In the introduction section, the development process of turbulence detection was introduced, but the differences between these development processes and this manuscript were not discussed. Please carefully refine them. For example, the methods used by Doppler LiDAR to detect turbulence, as described in references 18-22, and their differences and connections with this manuscript.**

Response: *As the reviewer suggests, we have modified the introduction section. The existing studies are based on the indirect acquisition of atmospheric turbulence parameters using structure function. It is called the structure function method, which rely on the theory of isotropy and consider the power-law exponent to be -5/3. We propose a method for directly measuring atmospheric turbulence parameters based on spectral analysis using coherent Doppler wind lidar. In our method, the power-law index is left as a free parameter. We have added the relevant content mentioned above to the text.*

**2. In section 2 of the manuscript, did you mention that your CSAT data is averaged over 30 minutes? Does this exceed the atmospheric turbulence freezing time?**

Response: *Thanks for the reviewer's comment. We have added the texts in revised version. The analysis of atmospheric boundary layer turbulence fluctuations, such as spectral analysis, is based on the assumption that atmospheric turbulence fluctuations are stationary. In order to ensure that the data meets this stationarity requirement within 30 minutes, we used turbulence stationarity coefficient in data processing, thereby ensuring that the observation time does not exceed the turbulence freezing time. We have added the relevant content mentioned above to the text.*

*The following content can be seen in the revised version:*
*"The analysis of atmospheric boundary layer turbulence fluctuations, such as correlation analysis and spectral analysis, is based on the assumption that atmospheric turbulence fluctuations are stationary. The actual atmospheric turbulence field is influenced by various factors and does not have stationarity characteristics (Massman, 2006). However, if a shorter observation time is used, under relatively stable weather conditions and flat underlying surface conditions, atmospheric turbulence can be approximated as static. Turbulence stationarity requires that the main statistical variables of turbulence remain stable within the observation time, that is, the mean of the variance of the entire time period within an observation time period is roughly equal to the mean of*

*the sum of variances of each period (Massman, 2006). In this study, data screening is conducted by determining whether the deviation between the mean variance within a 30 min observation time and the mean variance of six 5 min covariance samples within the same period is less than 0.3. Turbulence stationarity can be achieved through the stationarity coefficient, Δst, as shown in Equation (1) (Massman, 2006)*

$$\Delta st = \left|(\sigma^5 - \sigma^{30})\right| / \sigma^{30}$$

*where σ30 represents the variance of wind speed within 30 min and σ5 represents the average variance of six 5 min wind speed measurements made within 30 min. If Δst > 0.3, the data within these 30 minutes are discarded. The ratio of discarded data to all data used in this study is approximately 10%.*

**3. In the abstract section, it was explained that you have for the first published a spatiotemporal distribution map of the power law exponent of the inertial subrange, which shows the non-uniformity of atmospheric turbulence at different altitudes. I am not very clear about what I have read in sections 3.1 and 3.2 of your manuscript, because the new method you proposed is not expressed clearly. I can roughly understand it as using spectral analysis for fitting and formulating a system of equations to solve turbulent kinetic energy κ and the power-law exponent n (and n is not fixed -5/3). So it is important for you to carefully explain your new method and its differences from previous research methods.**

Response: *As the reviewer suggests, we have modified the introduction section and method section. The structure function method assumes isotropy in atmospheric turbulence and cannot obtain power-law exponents. By using wind lidar to obtain atmospheric turbulence spectra, not only can turbulent kinetic energy be directly obtained, but also power-law exponents can be obtained, which can be applicable to different atmospheric conditions and theoretically has higher applicability and accuracy. We have added the relevant content mentioned above to the text.*

**4. Turbulence has both spatial and temporal scales, and directly measuring the spatiotemporal distribution of turbulence has always been an important issue. And we know that most turbulence spectra are distributed between 0.1Hz and 100Hz. In section 3.2 of the manuscript, the highest frequencies of turbulence spectra that can be monitored by ultrasonic anemometers and wind LiDAR are 5 Hz and 0.1 Hz, respectively. Is this the reason for the significant difference in detection results between the two sensors? What is the contribution to the difference in detection results?**

Response: *As the reviewer suggests, we have modified the texts in revised version. Figure 2 shows a comparison of turbulence spectra between wind lidar and ultrasonic anemometer. From it, it can be seen that there is a very high consistency between the two in the overlapping region of the frequency range. In this study, in order to avoid differences caused by the size of the frequency domain, the frequency range was selected as the overlapping area of the two during the comparison process. So the difference in the highest frequency does not lead to a difference in detection results between the two sensors.*
*There are two factors that contribute to the difference in detection results. Firstly, as the installation of the ultrasonic anemometer on the north side of the gradient tower, when the wind direction approaches the south wind, there may be some differences between the two due to the obstruction of the gradient observation tower itself. Secondly, there is a difference in accuracy in vertical wind speed. The ultrasonic anemometer has a starting wind speed due to its measurement principle. Therefore, the data accuracy of the ultrasonic anemometer slightly decreased. The wind lidar directly measures vertical wind speed in principle, and there is no issue with starting wind speed. So there are some differences between the two. We have added the relevant content mentioned above to the text.*

**5. From your results, the comparison between LiDAR and CSAT is a single point comparison around 320 meters. How can the comparison of single points be close to prove that the spatial distribution of the results**

**detected by LiDAR is correct? Why didn't you compare a few more points below 320 meters in the vertical direction?**

Response: *Thanks for the reviewer's comment. Four three-dimensional ultrasonic anemometers are installed at the height of 10m, 40m, 160m, 320m on the gradient observation tower, respectively. The ultrasonic anemometers at heights of 10m and 40m are located within or close to the blind zone of the wind lidar, so we will not include them in the comparison. As the reviewer suggests, we added a comparison between the ultrasonic anemometer and the wind lidar at a height of 160m, and the results showed good consistency. We have added the relevant content mentioned above to the text.*

*The following content can be seen in the revised version:*

*Figure 10 shows a comparison of the long-term, continuous turbulent kinetic energy obtained from the wind lidar and three-dimensional ultrasonic anemometer from January 2nd to 10th, 2022, where panel (a)– (c) and (d)– (f) correspond to the heights of 160 m and 320 m, respectively. From the graph, we can see that the results of the wind lidar and ultrasonic anemometer are relatively consistent at the different heights. Figure 11 shows the correlation of the turbulent kinetic energy measured by the wind lidar and ultrasonic anemometer in all three directions, where panels (a)–(c) and (d)–(f) correspond to the same heights of 160 m and 320 m, respectively, with sample sizes (N) all greater than 2800. From the figure it can be seen that the results are relatively linear, with correlation coefficients greater than 0.9 and the slope of the fitting line approaches 1 in the U, V, and W directions. Compared to the U and V components, the consistency of the comparison results in the W direction is the highest. This is because under different horizontal wind directions, the results of the U and V components are more susceptible to interference from the gradient tower itself, while the vertical wind speed (W) is not affected by this, thus yielding better results. This comparison of the continuous results indicates that the proposed method has a high applicability and accuracy.*

[Figure]

**Figure 10.** Comparison of the turbulent kinetic energy obtained from the wind lidar and three-dimensional ultrasonic anemometer from January 2nd to 10th, 2022 in the *U*, *V*, and *W* directions at the heights of 160 m (a)–(c) and 320 m (d)–(f).

[Figure]

**Figure 11.** Correlation between the turbulent kinetic energy obtained with the wind lidar and three-dimensional ultrasonic anemometer in the *U*, *V*, and *W* directions at the heights of 160 m (a)–(c) and 320 m (d-(f).

**6. In the results and discussion of section 4, we believe that its detection ability will be limited for such a system and sampling frequency. For example, whether the consistency of turbulence is better only in clear skies (Figure 3a-c), and the consistency of results becomes worse in severe convective weather (Figure 7a-c).**

Response: *As the reviewer suggests, we have modified the texts in revised version. The wind lidar can work normally in sunny, cloudy, and other weather conditions, but it cannot work in rainy days. Figures 7 (a) - (c) show the data for September 11, 2022. On that day, there was a southeast wind. As the installation of the ultrasonic anemometer on the north side of the gradient tower, the ultrasonic anemometer on that day may be affected by the wind obstruction caused by gradient observation towers, and the difference between the two is significant. From the nighttime data of the ultrasonic anemometer, such as near 0 o'clock, its turbulent kinetic energy is close to its intensity during the day, it can be inferred that it is affected by the interference of the gradient observation tower itself. We have added the relevant content mentioned above to the text.*

**7. In Figure 9, we can clearly see that when the turbulent kinetic energy is high, the turbulent kinetic energy and power-law exponent of LiDAR and CSAT3 in the vertical direction are significantly different. Why is this? Does it mean that this method is flawed in cloudy weather conditions?**

Response: *As the reviewer suggests, we have modified the texts in revised version. From Figures 9 (a) - (c), it can be seen that the consistency of turbulent kinetic energy between the two is very high on this day. In Figures 9 (d) and (e), the power-law exponents of the two in the U and V directions are also relatively consistent. This means that the method proposed has good performance in turbulent kinetic energy and power-law exponent in the U and V directions in cloudy weather. For Figure 9 (f), there is a significant difference in the power-law exponent between the two in the W direction. Because it was a cloudy day, the vertical wind speed on that day was relatively*

*low, about 0.1 m/s. The ultrasonic anemometer has a starting wind speed (~0.01 m/s) due to its measurement principle. Therefore, the data accuracy of the ultrasonic anemometer slightly decreased on that day. The wind lidar directly measures vertical wind speed in principle, and there is no issue with starting wind speed. So there are some differences between the two. Therefore, this deviation is reasonable, indicating that our wind lidar can still operate under cloudy weather conditions. We have added the relevant content mentioned above to the text.*

**8. Some normative issues in drawing, such as: the area near the ground pointed by the red arrow in Figure 1a is white, while the area where the LiDAR is placed in the right image is clearly a green lawn. This is very confusing; Can the lines in Figures 3, 5, 7, 9, and 10 be distinguished by dotted lines, dashed lines, etc?**

Response: *Thanks for the reviewer's comment. Figure 1 (a) is a panoramic photo of the gradient observation tower when it was just built. At that time, the site under the gradient observation tower was not greened. The panoramic photos were not updated later. So we used this photo mainly to illustrate the environment around the gradient observation tower.*

*As the reviewer suggests, we have updated the Figures 3, 5, 7, 9, and 10 in the revised version.*

**9. At the end of section 3.1, turbulent kinetic energy κ was introduced. However, there is no mathematical relationship between it and the aforementioned formula, which is very confusing.**

Response: *Thanks for the reviewer's comment. We have added the Equation (7).*

*The following content can be seen in the revised version:*

*The slope can be obtained by performing linear fitting on x and log(Sz(f)), which yields the power-law exponent n. The turbulent kinetic energy, κ, in a certain frequency range [f0, f1] can be obtained by*

$$\kappa = \sum\nolimits_{f_0}^{f_1} S_z(f)$$

*When κ and n are known, the dissipation rate can be obtained from Equation (2). Therefore, this paper mainly discusses turbulent kinetic energy, κ, and the power-law exponent, n.*

We thank the Editor again for his helpful suggestions.

On behalf of all authors,
Sincerely,
Honglong Yang

Shenzhen National Climate Observatory
Meteorological Bureau of Shenzhen Municipality
518000 Shenzhen, China
E-mail:  yanghl01@163.com

---

## Author Comment (AC2)

<h1 style="text-align:center">AMT-2023-249- Response Letter</h1>

Dear Editor and reviewers,

We would like to thank the reviewers and editor for their comments that have allowed us to further clarify some aspects of the manuscript in this revised version. Hereafter, we report reviewers' comments and our replies (*in italics*). For yours and reviewers' convenience we have put the corresponding major changes introduced in red color in the revised version of the manuscript.

**Reviewer 2:**

The manuscript presents a method to directly determine two atmospheric turbulence parameters – turbulent kinetic energy (TKE) and power component – using measurements from a coherent Doppler wind lidar. Due to my expertise remit, I am unable to justify the novelty of the proposed approach; however, as an experienced academic writer in urban climate, I do find there is room for improvement in this manuscript before potential publication.

Specifically, could the authors please address the following concerns/questions:

1. **Clarify the term "atmospheric turbulence parameters" early on. There are numerous ATPs; for clarity in this work's context, use "power component" in the title and elaborate on this in the introduction.**

Response: *As the reviewer suggests, we have modified the title and the introduction section. The new title is "Directly Measuring the Power-law Exponent and Kinetic Energy of Atmospheric Turbulence Using Coherent Doppler Wind Lidar".*

2. **Figure 1 is not a topographic map - provide more convincing evidence to represent local terrain features.**

Response: *As the reviewer suggests, we have updated the Figure 1.*

*The following content can be seen in the revised version:*

[Figure]

**Figure 1.** Layout diagram (a), topographic map (b) of the surrounding area of the meteorological gradient observation tower, and installation diagram of the wind lidar below the tower (c).

3. **Table 1: Verify the units of sample frequency and temporal resolution of wind profile data.**

Response: *As the reviewer suggests, we have updated the Table 1.*

**4. Section 3.1 lacks essential details about how these parameters were derived from measurements:**

  **- The turbulence power spectrum S**

  **- Turbulent kinetic energy κ**

Response: *Thanks for the reviewer's comment. We have modified the texts and added the Equation (7) in revised version.*

*The following content can be seen in the revised version:*

*The slope can be obtained by performing linear fitting on x and log(Sz(f)), which yields the power-law exponent n. The turbulent kinetic energy, κ, in a certain frequency range [f0, f1] can be obtained by*

$$\kappa = \sum_{f_0}^{f_1} S_z(f)$$

*When κ and n are known, the dissipation rate can be obtained from Equation (2). Therefore, this paper mainly discusses turbulent kinetic energy, κ, and the power-law exponent, n.*

**5. In section 3.2, when comparing power spectra, justify why measurement heights for wind lidar differ from those used with anemometers.**

Response: *Thanks for the reviewer's comment. Due to the spatial resolution of wind lidar data being 30 meters, heights close to 320 meters are either 300 meters or 330 meters. Therefore, we compared the wind speed data of the wind lidar at a height of 330 meters with the ultrasonic anemometer data at a height of 320 meters. As the reviewer suggests, we have modified the texts in revised version.*

**6. For figure 2, employ a scatter plot to enhance clarity.**

Response: *As the reviewer suggests, we have added a scatter plot in the Figure 2 in the revised version. Figures 2 (d) - (f) provide the correlation coefficients (R) corresponding to Figures 2 (a) - (c), respectively. It can be seen that the correlation coefficients are greater than 0.9 in all three directions.*

*The following content can be seen in the revised version:*

[Figure]

**Figure 2.** Comparison of the turbulence spectra obtained with the wind lidar and the ultrasonic anemometer in three directions: (a) *U*, (b) *V*, and (c) *W*, and the corresponding correlations (d), (e), and (f).

**7.   Line 191: "2022 January 1" - Choose an appropriate date format in English.**

Response: *As the reviewer suggests, we have modified the texts in revised version. Other similar formats have also been updated.*

**8.    Line 210: Clarify what is meant by "due to a lack of light."**

Response: *Thanks for the reviewer's comment. At night, due to the weakening of solar radiation, the kinetic energy in the vertical direction is suppressed and the power-law exponent increases. As the reviewer suggests, we have modified the texts in revised version.*

**9. Figures 3 and 7:**

  **- Specify the temporal resolution of results displayed.**

  **- For panels d, e, and f, narrow down the y-axis range to highlight result variations more effectively.**

Response: *As the reviewer suggests, we have added an explanation of time resolution and updated the Figures 3, 5, 7 and 9 in the revised version.*

**10.  Figures 4, 6, and 8: State which instrument was used for taking measurements in each caption clearly.**

Response: *As the reviewer suggests, we have updated the captions of Figures 4, 6 and 8 in the revised version.*

**11.  Figure 11:**

**- Include both: a) number of data points b) linear regression analysis between two methodologies**

**- Omit "kinetic energy" from axis labels.**

**- Standardise plotting ranges on both x and y axes for consistency.**

Response: *As the reviewer suggests, we have updated Figure 11 and added sample size (N) and linear regression analysis in the revised version.*

*The following content can be seen in the revised version:*

[Figure]

**Figure 11.** Correlation between the turbulent kinetic energy obtained with the wind lidar and three-dimensional ultrasonic anemometer in the *U*, *V*, and *W* directions at the heights of 160 m (a)–(c) and 320 m (d-(f).

**12. In your conclusion section discuss any limitations in the developed approach.**

Response: *Thanks for the reviewer's comment. As the reviewer suggests, we have added a discussion on the limitations of the proposed method in the conclusion. The proposed method has some limitations. First, the method based on a spectral analysis that requires the atmospheric turbulence fluctuations to be stable. Next, wind lidar cannot operate during heavy rainfall or snowy weather conditions, so it cannot be guaranteed to be applicable at all times. In addition, due to the maximum observation frequency of 0.2 Hz for wind lidar, the observed scale range of vortex motion is limited. However, the results of this study also indicate that in the inertial subrange, turbulence spectra outside the frequency of 0.2 Hz can be obtained through fitting and extrapolation. We have added the relevant content mentioned above to the text.*

We thank the Editor again for his helpful suggestions.

On behalf of all authors,
Sincerely,
Honglong Yang

Shenzhen National Climate Observatory
Meteorological Bureau of Shenzhen Municipality
518000 Shenzhen, China
E-mail: yanghl01@163.com

---

## Referee Report (RR1)

Firstly, the research on "the spatiotemporal distribution map of the power-law exponent of the inertial sub range" proposed by the author team in the manuscript is worthy of recognition. Secondly, the author team has provided good solutions for the review comments. Here, I think there are some modifications and improvements that can better present this article to everyone. So my suggestion is that after minor revision, it can be published. The specific suggestions are as follows:

1. In the abstract section: I think these statements are worth discussing, "Here, we propose a method for …proving the superiority of our method.". Firstly, based on facts, explain what methods were proposed, what actions were taken, and what problems were solved. Please do not exaggerate excessively; Secondly, these few sentences are relatively long and can be difficult to understand, leading to confusion.

2. In the Instruments and Data Quality Control section: We all know that the definition of the boundary layer is the part of the troposphere directly affected by the ground, with a time scale of 1 hour. If you set the comparison sample time to 30 minutes here, I think many turbulence effects will be averaged out severely. The implication is that beyond the freezing time of atmospheric turbulence, we know that turbulence has a time period of 10 seconds to 10 minutes. The fluctuation of wind speed is also like this, which is described in detail in the reference you mentioned (Stull, R. B.1988.). I think the author's description here may still be a bit confusing. Although you compared the fluctuation between 30 minutes and 5 minutes with a difference of less than 0.3, is this fluctuation too large for turbulence? Implicitly, you don't have convincing evidence to suggest that this 0.3 has little effect on turbulence, and these are only under clear sky turbulence conditions. Also, will a 30-minutes average result in relatively small changes in turbulence? These require careful and rigorous thinking from the author and their team.

3. I don't quite agree with the analysis of Figures 9 and 10, mainly because the author directly summarizes this difference as: " This is because under different horizontal wind directions, the results of the U and V components are more susceptible to interference from the gradient tower itself, while the vertical wind speed (W) is not affected by this, thus yielding better results." I think this is not rigorous. Based on the comparison provided by the author (Figures 7 and 9), the turbulent kinetic energy is not 100% consistent. Does this mean that the method can also lead to errors? Is there any other possibility?

4. The author has an error in the abstract and conclusion that needs to be corrected. The long-term comparative observation in Figure 10 is no longer a month.

5. I have carefully reviewed the manuscript of the author team and I would like to raise a question here. When introducing the method, the author mistakenly treated the kinetic energy dissipation rate $\varepsilon$ as a fixed constant, which is incorrect. Usually, this method introduces errors, which are not easy to evaluate. This is when I see significant differences in turbulent flow energy in Figures 3, 7, and 9, the power-law exponent does not follow at this moment. Of course, this viewpoint does not affect the idea that this article is a good one, but we cannot ignore this issue. Please reflect the answer to this question in the theory, results and discussion, which will better reflect the author's scientific research attitude.

6. In addition, there are some normative technical issues: for example, the legend in Figure 9c only has wind lidar; For example, in Figure 10, the vertical coordinate units should be

kept as consistent as possible with the previous ones; The y-axis in Figures 3, 7, and 9 should be changed to "Power-law exponent".